# New *Cladosporium* Species from Normal and Galled Flowers of Lamiaceae

**DOI:** 10.3390/pathogens10030369

**Published:** 2021-03-19

**Authors:** Beata Zimowska, Andrea Becchimanzi, Ewa Dorota Krol, Agnieszka Furmanczyk, Konstanze Bensch, Rosario Nicoletti

**Affiliations:** 1Department of Plant Protection, University of Life Sciences, 20-068 Lublin, Poland; beata.zimowska@up.lublin.pl (B.Z.); ewa.krol@up.lublin.pl (E.D.K.); agnieszka.furmanczyk@up.lublin.pl (A.F.); 2Department of Agricultural Sciences, University of Naples Federico II, 80055 Portici, Italy; andrea.becchimanzi@unina.it; 3Westerdijk Fungal Biodiversity Institute, Uppsalalaan 8, 3584 CT Utrecht, The Netherlands; k.bensch@wi.knaw.nl; 4Council for Agricultural Research and Economics, Research Centre for Olive, Fruit and Citrus Crops, 81100 Caserta, Italy

**Keywords:** *Asphondylia* flower galls, *Cladosporium cladosporioides* species complex, *Cladosporium neapolitanum*, *Cladosporium polonicum*, Lamiaceae

## Abstract

A series of isolates of *Cladosporium* spp. were recovered in the course of a cooperative study on galls formed by midges of the genus *Asphondylia* (Diptera, Cecidomyidae) on several species of Lamiaceae. The finding of these fungi in both normal and galled flowers was taken as an indication that they do not have a definite relationship with the midges. Moreover, identification based on DNA sequencing showed that these isolates are taxonomically heterogeneous and belong to several species which are classified in two different species complexes. Two new species, *Cladosporium polonicum* and *Cladosporium neapolitanum*, were characterized within the *Cladosporium cladosporioides* species complex based on strains from Poland and Italy, respectively. Evidence concerning the possible existence of additional taxa within the collective species *C. cladosporioides* and *C. pseudocladosporioides* is discussed.

## 1. Introduction

Fungi belonging to the genus *Cladosporium* (Dothideomycetes, Cladosporiaceae) are ubiquitous and reported to be able to colonize a huge diversity of substrates in any natural or anthropized environment on earth [1]. They are well known as plant disease agents [1,2,3,4,5], but also reported as pathogens of animals [6] and humans [7,8,9], and are considered among the most widespread fungi in buildings and indoor environments [10]. Other species are endophytic or have been reported from soil, dung or leaf litter [11,12,13,14]. Recent investigations have shown that pathogenic strains usually belong to species mostly known as saprophytes, underlining the importance of an accurate assessment of the phylogenetic relationships for the identification of specialized lineages and possible cryptic species [1,7,8,10]. In fact, classification based on morphology has proved to be problematic due to the infrequency of the perfect stage and the absence of outstanding differences in the conidial structures, so that culturing and microscope observations only allowed a partial separation of taxa possibly representing collective species. Therefore, widespread species, such as *C. cladosporioides, C. herbarum*, and *C. sphaerospermum*, are now regarded as species complexes (s.c.), disclosing a broader variation as and when the characterization of new strains proceeds from new ecological contexts and geographic areas [3]. Indeed, the easier access to DNA sequencing and online databases is ongoingly supporting the distinction of novel species, to such an extent that within the *C. cladosporioides* s.c. their number has currently raised up to 67 (Table 1).

One of the most fruitful investigational fields on the occurrence of *Cladosporium* species is represented by the ecological interactions with other organisms. Particularly, many studies have evidenced the ability of these fungi to exert antagonistic effects against pests and pathogens of crops, which is also supported by their widespread association with plants as epiphytes or endophytes [15,16]. However, in other cases it has not been clearly established whether the finding of *Cladosporium* is related to a definite symbiotic interaction or to a merely saprophytic condition. One of these cases is represented by the association with midges belonging to the Asphondyliinae (Diptera, Cecidomyidae), which on many plant species induce the formation of galls where larvae develop by feeding on a mycelial mat lining the gall walls. The fungal counterpart was identified as *Cladosporium* sp. in early investigations on these peculiar symbiotic associations [17,18]. Later on, this role was questioned, although occurrence of *Cladosporium* in galls was confirmed in recent reports from several plants and countries worldwide [19,20,21]. Moreover, the finding of *Cladosporium* conidia in mycangia and on the body surface of egg-laying midges could possibly support the conjecture that the insect may actively spread the fungus during oviposition [21,22].

In the course of a cooperative investigational activity on the fungal associates developing in galls produced by midges of the genus *Asphondylia* in flowers of several species of Lamiaceae [23,24], strains of *Cladosporium* were frequently recovered during the isolation attempts. However, unlike *Botryosphaeria dothidea* which was only isolated from galls [25], *Cladosporium* isolates were also obtained from the inner parts of normal flowers and from achenes, indicating that their presence in the flower microenvironment is independent by the insect, and is likely to not affect flower physiology. In the absence of previous assessments, identification at species level appeared to be fundamental in order to conclusively establish whether these isolates are taxonomically homogeneous, hence, to be possibly regarded as specialized gall associates, or rather occur as unrelated saprophytes.

## 2. Results

### 2.1. Cladosporium Isolates

As discussed above, *Cladosporium* strains were quite frequently isolated from flower galls, their inquiline insects and normal flowers collected on several Lamiaceae species examined in our investigation. Forty strains from the resulting collection were selected to be examined in this study (Table 2). The list included representatives from all the sampled plant species, with a prevalence of isolates from galls depending on the higher number of isolations which were performed from this source.

### 2.2. Phylogenetic Analysis

Considering that recent work and revisions on the taxonomy of *Cladosporium* agree on the insufficient reliability of morphological characters for a correct species ascription [1,8,10], the selected isolates were directly processed for DNA extraction and sequencing of ITS, TEF1 and ACT regions. All the obtained DNA sequences listed in Table 2 have been deposited in GenBank, to be available for further taxonomic assessments.

Identification based on DNA sequencing and BLAST searches in GenBank showed that four strains could be ascribed to *C. ramotenellum* and three to *C. allicinum*, which are widespread saprobes within the *C. herbarum* s.c. [1]. However, the majority of the collected strains (33) were found to belong in the *C. cladosporioides* s.c., and were further analyzed for assessing their phylogenetic relationships with all the members of this taxonomic group. Overall, the phylogenetic analysis included 141 strains and was based on a nucleotide set of 1164 bp (536 bp for ITS, 373 bp for *TEF1*, and 255 bp for *ACT*).

The phylogram obtained through maximum likelihood (ML) analysis (Figure 1) shows that two strains from gall walls of *Asphondylia nepetae* can be respectively ascribed to *C. delicatulum* and *C. perangustum*, both known as saprobic and widely distributed species, while the remaining are grouped in four highlighted clades of the tree (A–D). The 16 isolates in group A are phylogenetically closely related to *C. pseudocladosporioides* (ML bootstrap/MP bootstrap/posterior probabilities = 92/88/1.0), with some exceptions. In particular, isolate Cl3 appears to be more closely related to *C. crousii*, while Th/K/258 and Th/lg/2334, two Polish isolates from *Thymus vulgaris,* form an independent clade (ML bootstrap/MP bootstrap/posterior probabilities = 87/87/1.0). The nine isolates in group B are phylogenetically closely related to *C. cladosporioides* (ML bootstrap/MP bootstrap/posterior probabilities = 96/82/1.0), with a certain degree of variation which is inferable from their distribution in several subclades. The two isolates in group C, collected in Campania from *Micromeria graeca*, form an independent clade in proximity to *C. xylophilum* (ML bootstrap/MP bootstrap/posterior probabilities = 100/100/1.0). Finally, the four isolates in group D result to belong to the same clade as *C. europaeum* (ML bootstrap/MP bootstrap/posterior probabilities = 100/100/1.0).

### 2.3. Species Delimitation Assay

These four different groupings were separately analyzed along with the reference strains of the most closely related species by means of the automatic barcode gap discovery (ABGD) and general mixed Yule-coalescent (GMYC) methods for species delimitation. Results of these supplementary analyses were in agreement with one another, increasing the confidence of taxonomy assignments.

In detail, the set of strains forming group A was integrated by a reference strain for each of the six candidate species (*Cladosporium* sp. 3–8) which have been recently pointed out to exist within the *C. pseudocladosporioides* aggregate [7,8]. Besides the outgroup strain (*C. hillianum*), this overall set segregated in 13 taxa (Figure 2). The largest one is represented by 13 isolates, from both countries and miscellaneous origins, in association with the canonical reference strains of *C. pseudocladosporioides*, thereby confirming their belonging to this species *sensu stricto*. Moreover, an Italian isolate from *Clinopodium vulgare* (Cl3) matches with the representative of “*Cladosporium* sp. 5” (UTHSC DI-13-245), while two Polish isolates from *T. vulgaris* form an independent clade, which is close but separate from “*Cladosporium* sp. 7” (UTHSC DI-13-218).

Likewise, considering the variation resulting in the general phylogenetic analysis, the set of strains in group B was integrated with six additional representatives of *C. cladosporioides* of which sequences are available in GenBank (Figure 3). In this case, the congruent analyses based on ABGD and GMYC indicate that 9 isolates of assorted origin in our sample are differentiated in 4 groups, each including at least one reference strain of *C. cladosporioides*.

Finally, the species delimitation analysis carried out for group C (Figure 4) indicates, with strong support, that the two isolates from receptacles of *M. graeca* represent an independent species, close to *C. xylophilum*, while the two couples of isolates from both countries included in group D cluster together with *C. europaeum*, confirming their ascription to this species (Figure 5).

### 2.4. Morphological Characteristics

Isolates showing significant divergence from known species of the *C. cladosporioides* s.c., thereby representing candidate novel species as revealed by the above mentioned phylogenetic and species delimitation analyses, were further examined with reference to their morphological and cultural characteristics. Morphological characters were evaluated in comparison to the phylogenetically most closely related species (Table 3). For the species related to *C. xylophilum*, some differences were observed consisting in shorter conidiophores and ramoconidia, a lower number of hila on the secondary ramoconidia, and slower growth on all culturing media. Based on evidence gathered from phylogenetic analyses and morphological examination, these candidate species are described as follows.

### 2.5. Taxonomy

*Cladosporium polonicum* Zimowska & Król *sp. nov.*—MycoBank MB839011; Figure 6.

Similar to *C. pseudocladosporioides*, from which it differs in forming slightly shorter, 0−1 septate ramoconidia and shorter secondary ramoconidia.

Etymology: Named after the country where the representative strains were collected, Poland.

Mycelium immersed and superficial, hyphae unbranched or sparingly branched, up to 4 µm wide, septate, sometimes constricted at septa, subhyaline to pale olivaceous-brown, smooth or almost so, walls sometimes slightly thickened, sometimes irregular in outline due to swellings and constrictions, cells sometimes swollen, fertile hyphae minutely verruculose, mainly at the base of conidiophores. Conidiophores macronematous, sometimes also micronematous, solitary or in small loose groups, arising terminally and laterally from hyphae, erect, straight to slightly flexuous, cylindrical-oblong, non nodulose, sometimes once geniculate-sinuous or slightly swollen at the apex, unbranched or once branched, occasionally three times, branches often only as short denticle-like lateral outgrowth just below a septum, sometimes attenuated towards the apex, 0−5 septate, sometimes slightly constricted at septa, pale to pale medium olivaceous-brown, sometimes paler towards the apex, smooth or almost so, or asperulate or finely verruculose, walls slightly thickened or unthickened: (22−)42.7−151 × 2−3.6(−5.1) µm. Micronematous conidiophores filiform, narrower, not attenuated. Conidiogenous cells narrow, with 1−5 loci crowded at the apex, subdenticulate, 1−1.8 µm diam. Ramoconidia cylindrical-oblong, 0−1 septate pale olivaceous-brown, smooth, base broadly truncated, 2−3 µm wide, unthickened or slightly thickened, sometimes slightly refractive: 12.9−39.8 × 2.4−5.2 µm (av. 20.5 × 3.3). Secondary ramoconidia ellipsoid-ovoid to subcylindrical or cylindrical-oblong, 0−1(−3) septate, septum medium or often somewhat in the lower half, pale olivaceous to pale olivaceous-brown, smooth or almost so, sometimes slightly rough-walled, walls unthickened, with 1−4 distal hila, conspicuous, subdenticulate, somewhat thickened: 7.9−23.2 × 2.4−4 µm (av. 12.3 × 3.1). Microcyclic conidiogenesis not observed. Conidia very numerous, catenate, in branched chains, branching in all directions. Small terminal conidia obovoid, ovoid to limoniform or ellipsoid, sometimes subglobose, apex rounded or attenuated towards apex and base, 3−5.8 × (1.5−)2−3 µm, av. 5 × 2.2. Intercalary conidia ovoid, limoniform to ellipsoid or subcylindrical, 0(−1) septate, slightly attenuated towards apex and base, with 1−4 distal hila: 5.5−11 × 2−3.9 µm, av. 7.8 × 2.7.

Culture characteristics: Colonies on PDA attaining 63−70 mm diam after 14 days at 25 °C, olivaceous-grey, to grey olivaceous. Reverse leaden grey to olivaceous-black, felty-floccose, concentric ring visible in the center of colony, margin white very narrow up to 2 mm, glabrous, regular, aerial mycelium velvety to felty, growth flat, without exudates formed, sporulation profuse. Colonies on malt-extract agar (MEA) reaching 53−58 mm, grey olivaceous, reverse iron grey, floccose, margin white very narrow up to 2 mm, regular, glabrous, aerial mycelium velvety to felty, growth flat, without exudates, sporulation profuse, concentric ring visible in the center of colony. Colonies on oatmeal agar (OA) attaining 55−60 mm, olivaceous to grey olivaceous or olivaceous-buff, pale olivaceous-grey to greenish-grey towards margins. Reverse pale greenish-grey, leaden grey to iron grey, floccose, margin colorless up to 2 mm, glabrous, regular, aerial mycelium floccose to felty, radial sectors in the center of colony, sporulation profuse.

Specimens examined: POLAND, Lubelskie voivodeship, Fajsławice, from gall of *Asphondylia serpylli* Kieffer on thyme (*Thymus vulgaris* L., Lamiaceae), B. Zimowska, 18 June 2016, Th/lg/2334, holotype, preserved in a metabolically inactive state at the mycological collection of the Department of Plant Protection of the University of Life Sciences in Lublin; Konopnica, from receptacle in flower of catnip (*Nepeta cataria* L., Lamiaceae), B. Zimowska, 28 June 2018, Th/k/258.

Notes: Evidence resulting in a recent study [10] indicated *C. crousii* to be probably conspecific with *C. pseudocladosporioides*, also with reference to the very close species descriptions. Conversely, in our analysis, the species delimitation methods support both *C. crousii* and *C. polonicum* as separate species.

*Cladosporium neapolitanum* Zimowska, Nicoletti & Król *sp. nov.*—MycoBank MB839012; Figure 7.

Similar to *C. xylophilum*, from which it differs in forming shorter conidiophores, shorter ramoconidia and secondary ramoconidia, and for a lower number of hila at the apex of secondary ramoconidia.

Etymology: Named after the city of Napoli, Italy, in which surroundings the representative strains were collected.

Mycelium immersed and superficial, hyphae unbranched or loosely branched, 1−4(5) µm wide, septate, not constricted at septa, sometimes with irregular swellings and outgrowths, subhyaline to pale or medium olivaceous-brown, smooth to asperulate, minutely verruculose or irregularly verrucose, and rough-walled, with wart-like structures on the surface, walls unthickened, occasionally swollen at the base of conidiophores. Conidiophores macro- to sometimes micronematous, solitary, arising terminally and laterally from hyphae, erect, straight to slightly flexuous, cylindrical-oblong, usually neither nodulose nor geniculate, sometimes subnodulose at the uppermost apex, occasionally geniculate-sinuous, unbranched, sometimes once branched, 0−5 septate, sometimes slightly constricted at septa, pale to medium olivaceous-brown, smooth or almost so, sometimes somewhat irregularly rough-walled or verruculose, especially towards the base, sometimes wider at the base, or slightly toward the apex, walls slightly thickened: (28.1−)44.4−142.5 × 2.4−4.2 µm; growth sometimes proceeding at an angle 45−90°. Micronematous conidiophores paler, subhyaline to pale-olivaceous-brown, smooth or almost so. Conidiogenous cells terminal and intercalary, loci crowded at the apex forming clusters of pronounced scars, 2−5 conidiogenous loci formed at about the same level, loci often situated at lateral shoulders due to sympodial proliferation, loci 1−2 µm diam. Ramoconidia occasionally formed, cylindrical-oblong, 0(−1) septate, smooth, base broadly truncate 10.1−22.2 × 2.2−3.7(−4.3) µm. Secondary ramoconidia ellipsoid to cylindrical-oblong or irregular in outline, 0−1(−3) septate, septum median or somewhat in the upper half, not constricted, with 2−4 distal hila, crowded at the apex or situated on small lateral prolongations, pale olivaceous to pale medium olivaceous-brown, smooth or almost so, walls unthickened or almost so, hila conspicuous, subdenticulate to denticulate, (5.7−)7.4−16.6 × (1.4−)1.8−3.1 µm, av. 10 × 2.4. Conidia numerous, catenate, in densely branched chains, branching in all directions, straight. Small terminal conidia subglobose, obovoid, sometimes globose, aseptate, slightly attenuated towards apex and base, apex broadly rounded. Small terminal conidia and intercalary conidia almost smooth to often irregularly rough-walled, loosely verruculose to verrucose, attenuated towards apex and base, (1.7−)2.2−4.9 × 1.6−2.5(−2.8) µm, av. 3.5 × 2. Intercalary conidia ovoid, limoniform to ellipsoid or subcylindrical, sometimes irregular in outline, especially towards the distal end, due to numerous hila arranged in sympodial clusters of pronounced scars, 0−1 septate, septum median, not constricted, 5.8−10.5 × 2−3.3 µm, av. 7.7 × 2.5.

Culture characteristics: Colonies on PDA attaining 47 mm diameter after 14 days, pale green. Reverse iron-gray to brown-black, floccose to fluffy, margin white narrow up to 2−3 mm, slightly irregular, aerial mycelium abundant, velvety to floccose, loose to dense, growth flat, radial sectors visible in the center of colony, without exudates, sporulation profuse. Colonies on MEA reaching 37−40 mm, pale green. Reverse olivaceous to iron-black, velvety to floccose-felty, margin white narrow up to 2−3 mm, slightly irregular, aerial mycelium abundant, velvety to floccose, loose to dense, growth flat, radial sectors visible in the center of colony, without exudates, sporulation profuse. Colonies on OA reaching 45−46 mm, pale green paler in the center. Reverse iron-black, velvety to floccose-felty, margin white narrow up to 2−3 mm, slightly irregular, aerial mycelium abundant, velvety to floccose, loose, growth flat, radial sectors visible in the center of colony, without exudates, sporulation profuse.

Specimens examined: ITALY, Campania region, Pontone, from receptacle in flower of *Micromeria graeca* (L.) Benth. ex Rchb. (Lamiaceae), R. Nicoletti, 9 Apr. 2016, MgPo1, holotype, preserved in a metabolically inactive state at the mycological collection of the Department of Plant Protection of the University of Life Sciences in Lublin; isle of Vivara, from receptacle in flower of *Micromeria graeca* (L.) Benth. ex Rchb. (Lamiaceae), R. Nicoletti, 3 June 2016, MgVi3.

## 3. Discussion

Despite *Cladosporium* having been quite frequently reported as an associate in galls formed by Asphondyliinae on many plant species, so far, no attempts have been done to perform identification at the species level in order to ascertain whether or not these findings are to be referred to a definite species. In fact, symbiotic associations generally involve specific adaptations by the symbionts which are considered to characterize single or closely related taxa. Confirming this assumption, *B. dothidea* is now regarded as the fungal associate of these midges after some controversies occurred in the past which in most instances derived from nomenclatural reassessments [25,26]. This evidence obviously contrasts the hypothesis that *Cladosporium* may have a role in this peculiar biological association. Observations reported in the present study reinforce this conclusion, with reference to the degree of diversity which has been pointed out in the pool of *Cladosporia* recovered from galled and non-galled flowers of some species of Lamiaceae. In fact, our investigation carried out in two geographically distant areas demonstrated that (i) isolates of the same *Cladosporium* species can be recovered from both galled and non-galled flowers, and (ii) isolates from galls can be ascribed to at least seven species belonging to two species complexes.

Considering the uneven sampling with reference to both the plant species and the geographic areas, no definite association can be inferred. Within the *C. herbarum* s.c., the three strains of *C. allicinum* were all recovered from flower receptacles of *Lamium* and *Lamiastrum* spp., while the four strains of *C. ramotenellum* were found in flower receptacle of *M. graeca* and in galls of *A. nepetae*. Both these species are reported to be of worldwide occurrence in association with many heterogeneous plants [1].

Within the *C. cladosporioides* s.c., the species *C. perangustum* and *C. delicatulum*, both represented by single isolates from gall walls of *A. nepetae*, are known to be saprobic and widely distributed [27]. Two couples of isolates of *C. europaeum* were found on different species of Lamiaceae in Poland and Italy, which is to be taken as an indication of a more widespread occurrence in Europe of this species supporting its appropriately chosen name. In fact, it was recently separated from *C. cladosporioides* based on isolates from miscellaneous plant materials and indoor environments collected in Denmark, Germany, Portugal, and the Netherlands [10]; from the latter country, it has also been recovered from brown algae (*Fucus* sp.) [28]. With, respectively, 9 and 13 strains of assorted origin from both countries, *C. cladosporioides* and *C. pseudocladosporioides* are confirmed to be the most common representatives of this species complex. They both also show a notable degree of genetic variation, which is indicative of the possible existence of cryptic species, as predicted in previous studies and revisions [3,4,7,8,10]. In this respect, our phylogenetic study demonstrated correspondence of a strain from galls collected on *C. vulgare* to one of the candidate taxa, ‘*Cladosporium* sp. 5′, defined in the study by Sandoval-Denis et al. [7]. This strain and the two representatives of *C. polonicum*, which are also in phylogenetic proximity with *C. pseudocladosporioides*, might have been mistakenly ascribed to the latter species if the use of DNA sequences had been limited to a BLAST searches in the GenBank database. Hence, it is clear that in the absence of reliable morphological characters, the use of sequence-based statistical methods able to assess the significant phylogenetic distances is to be recommended in view of a correct classification, as well as to avoid the accumulation of misleading identifications of strains which have DNA sequences deposited in public repositories.

Even if displaying a certain degree of variation, many strains fitted in the cluster of *C. cladosporioides*. Our species delimitation analysis indicate that this grouping could be differentiated in four species, each including at least one strain from our sample and one reference strain, supporting the expectation that more new species could be separated within the currently defined *C. cladosporioides.* Considering that a more resolutive analysis should include most of the over 100 strains whose complete sets of sequences are available in GenBank, we decided not to try to get to more conclusive assessments in the present work, and to provisionally confirm identification of these Lamiaceae strains as *C. cladosporioides*.

With reference to the description of *C. neapolitanum*, it is interesting to consider that in their fundamental revision Bensch et al. [3] pointed out the existence of a certain degree of variation within *C. xylophilum*, and that the possible existence of cryptic species would have required to be ascertained based on a broader strain sample. Our finding seems to represent the first occasion meeting this expectation.

Besides emphasizing the need of a thorough revision of strains currently classified as *C. cladosporioides* and *C. pseudocladosporioides*, by the finding of two novel species our study confirms the taxonomic heterogeneity of the *Cladosporium* complex associated with flowers of Lamiaceae. Indeed, these plants represent a fruitful investigational ground for studying diversity of these ubiquitous fungi, also with reference to the possible contribution by endophytic strains to the biosynthesis of components of essential oils and other bioactive compounds, representing the basic property sustaining their industrial exploitation [29,30,31,32]. Interestingly, two isolates of each new species were found in two ecologically homogeneous areas in Poland and in Italy. Future investigations will disclose if they should be regarded as regional entities, or rather as more widespread taxa.

## 4. Materials and Methods

### 4.1. Isolates Collection

*Cladosporium* isolates considered in this study (Table 2) were recovered over 4 years (2015–2018) from several Lamiaceae species. Particularly, this sampling activity involved cropped *T. vulgaris* and a stand of *N. cataria* in Lubelskie voivodeship, south-eastern Poland, and species of *Clinopodium, Micromeria*, and a few additional taxa from several locations in Campania and Basilicata regions, southern Italy. A single isolate recovered from *C. vulgare* collected in Grunau im Almtal, Austria, was also included. *Asphondylia* galls were only found on *Clinopodium nepeta, C. vulgare, Micromeria fruticulosa, M. graeca* in Italy, and *T. vulgaris* in Poland, which implies that the isolates from the other species were all obtained from normal flowers. Isolation of fungal associates from gall walls and inquilines, that is midge larvae or their parasitoids, was carried out as specified in previous papers [23,24]. Isolations from the inner flower parts (receptacle, ovaries, or achenes developing inside the flower calyx) were carried out on potato-dextrose agar (PDA: Difco, Paris, France) amended with streptomycin sulphate (200 mg L^−1^), after dissecting the flowers with a sterilized scalpel in a laminar flow hood. All isolates were transferred in pure culture for taxonomic identification and storage in our in-house mycological collections.

### 4.2. DNA Isolation, Amplification and Sequencing

Selected strains were sampled from the surface of PDA cultures with a scalpel. The mycelial matter was transferred to 1.5 mL Eppendorf tubes for DNA extraction. DNA isolation was performed by means of a DNA easy plant and fungi isolation kit (EurX, Gdańsk, Poland), according to manufacturer’s protocol. DNA concentration was estimated on 1.5% agarose gel, compared with GeneRulerTM DNA Ladder Plus (Thermo Scientific, Waltham, MA, USA), and measured through a NanoDrop 2000 spectrophotometer (Thermo Scientific). DNA samples were diluted to a concentration of 20 ng µL^−1^ and stored at −20 °C. Amplification of loci currently considered in taxonomy of *Cladosporium* [4] was carried out using primers ITS1 and ITS4 for the rDNA-ITS region, primers EF1-728F and EF1-986R for the translation elongation factor 1-alpha (TEF1) region, primers ACT-512F and ACT-783R for the actin gene (ACT) [33]. PCR reaction mixtures, containing 20 ng of genomic DNA, 0.2 mM dNTP, 0.2 mM of each primer, 10 × Taq buffer (10 mM Tris-HCl, 1.5 mM MgCl_2_, and 50 mM KCl), and 1 U of Taq polymerase, were adjusted to a final volume of 25 µL with sterile distilled water. PCR was conducted in a Biometra T1 thermocycler (Analytik Jena, Jena, Germany). The following reaction profile was applied: 95 °C—5 min, 35 cycles (95 °C—45 s, 52 °C—45 s, and 72 °C—45 s), with final elongation at 72 °C—5 min. PCR products were separated in 1.5% agarose gels containing ethidium bromide in Tris/borate/EDTA buffer, at 140 V, for 1 h. After checking and determining the size of the resulting PCR products, samples were submitted for sequencing to Genomed (Warsaw, Poland).

### 4.3. Phylogenetic Analyses

The obtained nucleotide sequences were blasted in GenBank for a provisional species identification. Moreover, sequences of isolates belonging to the *C. cladosporioides* s.c. were submitted to a phylogenetic analysis including GenBank sequences of one or two strains for all the described species in this s.c. (Table 1). Strain CPC 14300 of *C. ramotenellum,* a species belonging to the *C. herbarum* s.c., was used as outgroup. The combined ITS, TEF1, and ACT sequences were aligned by using Muscle [34] and manually adjusted with AliView software [35], where necessary. Congruence between the different datasets was tested through the partition homogeneity test in PAUP software version 4.0b10 [36]. Gaps were treated as missing characters. The phylogenetic analyses were carried out in conformity with recent protocols [7,37]. The best nucleotide substitution model (generalized time-reversible model with gamma distribution and a portion of invariable sites (GTR+G+I) for the three independent data sets) was estimated using jModelTest version 2.3 [38] following the Akaike criterion. Phylogenetic analyses of the concatenated sequence data for maximum likelihood (ML) were performed by using RAxML software version 8.2.12 [39] with the GTR+G+I model of nucleotide substitution and 1000 bootstrap replications. Concatenated sequences were also analyzed for maximum parsimony (MP) by using PAUP, under the heuristic search parameters with tree bisection reconnection branch swapping, 100 random sequence additions, maxtrees set up to 1000, and 1000 bootstrap. Bayesian analyses were done with a Markov chain Monte Carlo (MCMC) coalescent approach implemented in BEAST 2 [40], using the uncorrelated lognormal relaxed clock, the GTR+G+I model, and a coalescent tree prior. Bayesian MCMC was run for 50 million generations, and trees and parameters were sampled every 1000 generations. The resulting log files were entered in Tracer v1.6.0 to check trace plots for convergence and effective sample size (ESS). Burn-in was adjusted to achieve ESS values of ≥ 200 for the majority of the sampled parameters. While removing a portion of each run as burn-in, log files and trees files were combined in LogCombiner. TreeAnnotator was used to generate consensus trees with 25% burn-in and to infer the maximum clade credibility tree, with the highest product of individual clade posterior probabilities. Phylogenetic trees were drawn by using FigTree software (tree.bio.ed.ac.uk/software/figtree/) (accessed on 10 December 2020). Both the alignments and the trees were deposited in Zenodo.

### 4.4. DNA-Based Species Delimitation

Four clades of the tree resulting from the general phylogenetic analysis were selected for DNA-based species delimitation analysis, in order to provide taxonomic assignment for our isolates. We explored two different delimitation methods, the automatic barcode gap discovery (ABGD) [41] and the general mixed Yule-coalescent (GMYC) model [42]. These methods are among the most popular approaches for species delimitation based on sequence data and are frequently used in studies on fungal diversity [43,44,45]. When several methods for species delimitation offer congruent estimates of species diversity, the confidence of taxonomy assignment for a given dataset increases [44]. The ABGD method was tested through a web interface (abgd web, bioinfo.mnhn.fr/abi/public/abgd/abgdweb.html) (accessed on 15 December 2020). Before analysis, the model criteria were set as follows: variability (P) between 0.001 (Pmin) and 0.1 (Pmax), minimum gap width (X) of 0.1, Kimura-2-parameters and 50 screening steps. To perform the GMYC delimitation method, an ultrametric tree was constructed in BEAST 2, as described above. After removing 25% of the trees as burn-in, the remaining trees were used to generate a single summarized tree in TreeAnnotator v.2.0.2 (part of the BEAST v.2.0.2 package) as an input file for GMYC analyses. The GMYC analyses with a single threshold model were performed in R (R Development Core Team, www.R-project.org) (accessed on 15 December 2020) under the “splits” package using the “gmyc” function (R-Forge, r-forge.r-project.org/projects/splits/) (accessed on 15 December 2020).

### 4.5. Morphological Observations

Morphological observations were carried out for strains representing candidate novel species. For the assessment of cultural characteristics, the isolates were grown on PDA, oatmeal agar (OA), and malt-extract agar (MEA, Difco), for 14 days at 24 °C in the dark. The colony right and reverse colours were rated according to the charts set up by Rayner [46]. Micromorphological observations were made from colonies grown for 7 days at 24 °C on synthetic nutrient-poor agar (SNA). Squares of transparent adhesive tape (Dalpo, Poznań, Poland) were placed on the sporulating areas at the colony margin. Observations were carried out under a BA 210 microscope (Motic, Xiamen, China), and images were taken through a 1 MP Motic camera and Scopelmage 9.0 software (Bioimager, Vaughan, Canada). From each isolate, minimum, maximum and mean values were measured for a set of relevant characters considered in taxonomy of this fungal genus [3]. Descriptions followed terminology used in Bensch et al. [1].

## Figures and Tables

**Figure 1 pathogens-10-00369-f001:**
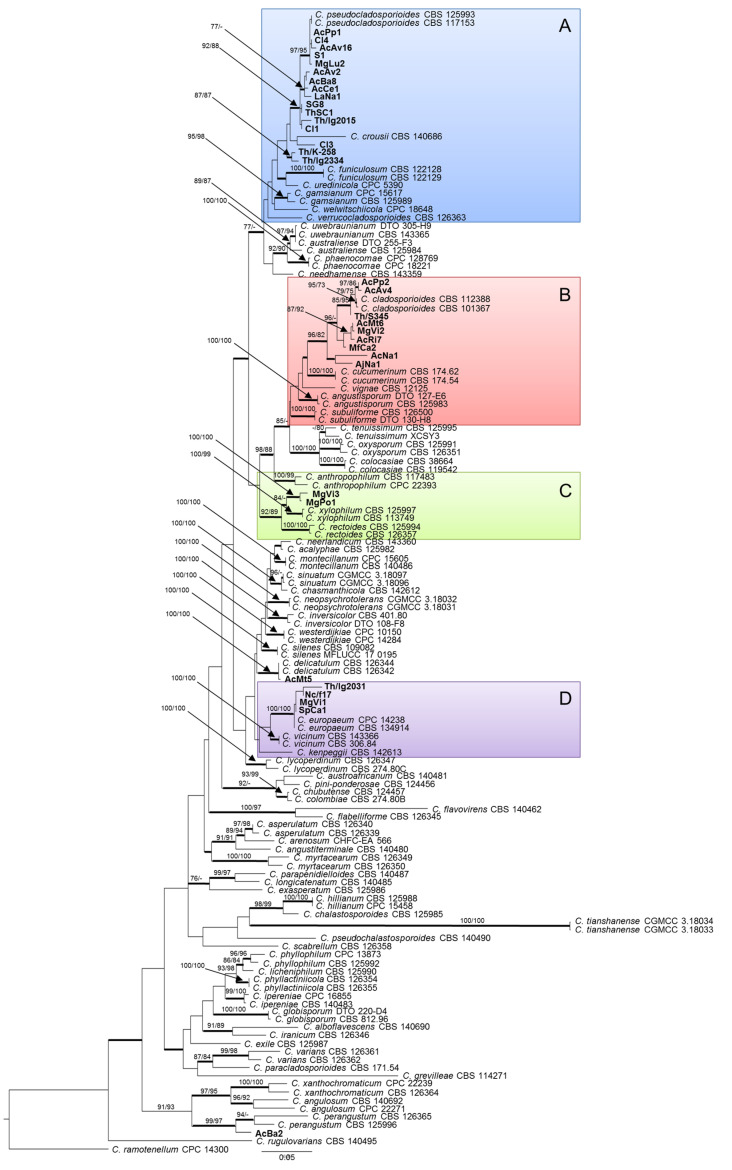
Phylogenetic tree based on maximum likelihood (ML) analysis of combined ITS, TEF1, and ACT sequences of 141 strains from the *C. cladosporioides* complex. Bootstrap support values ≥70% for ML and maximum parsimony (MP) are presented above branches as follows: ML/MP; bootstrap values <70% are marked with ‘-‘. Branches in bold are supported by Bayesian analysis (posterior probability >0.95). *C. ramotenellum* CPC 14300 was used as outgroup reference. Main clades are indicated by colored boxes A, B, C, and D.

**Figure 2 pathogens-10-00369-f002:**
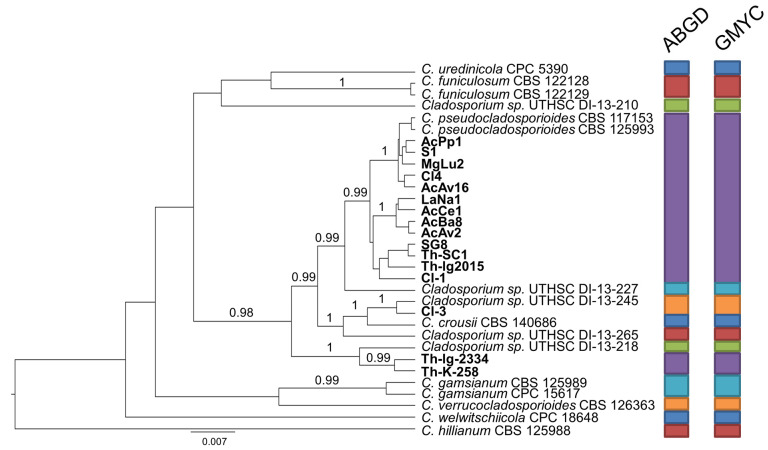
Ultrametric tree phylogeny of group A showing results of sequence-based species delimitation methods. The tree is the result of a Bayesian analysis performed in BEAST on the concatenated ITS, TEF1, ACT dataset. For each node, posterior probabilities (if >0.90) are presented above the branch leading to that node. Results of species delimitation analyses through automatic barcode gap discovery (ABGD) and general mixed Yule-coalescent (GMYC) methods are congruent, as visualized by colored boxes to the right.

**Figure 3 pathogens-10-00369-f003:**
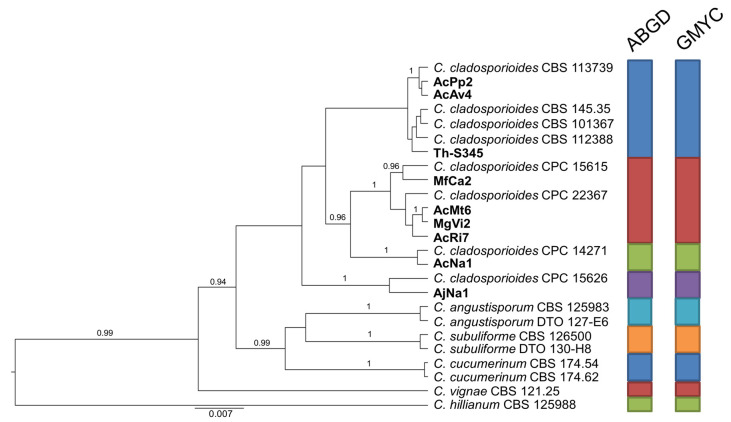
Ultrametric tree phylogeny of group B showing results of sequence-based species delimitation methods. The tree is the result of a Bayesian analysis performed in BEAST on the concatenated ITS, TEF1, ACT dataset. For each node, posterior probabilities (if >0.90) are presented above the branch leading to that node. Results of species delimitation analyses through ABGD and GMYC methods are congruent, as visualized by colored boxes to the right.

**Figure 4 pathogens-10-00369-f004:**
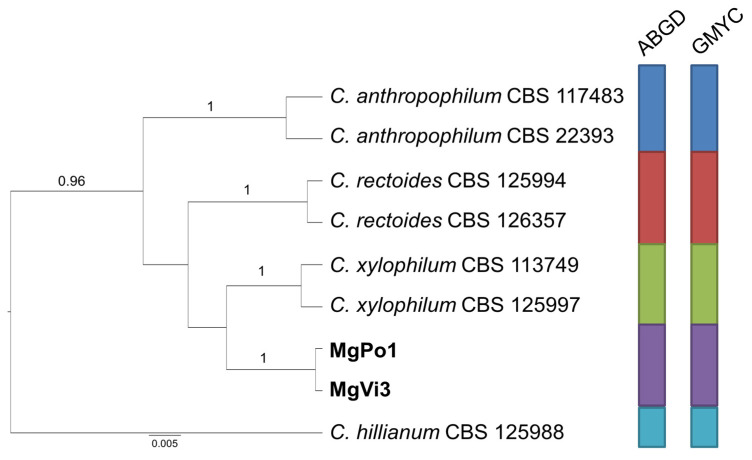
Ultrametric tree phylogeny of group C showing results of sequence-based species delimitation methods. The tree is the result of a Bayesian analysis performed in BEAST on the concatenated ITS, TEF1, ACT dataset. For each node, posterior probabilities (if >0.90) are presented above the branch leading to that node. Results of species delimitation analyses through ABGD and GMYC methods are congruent, as visualized by colored boxes to the right.

**Figure 5 pathogens-10-00369-f005:**
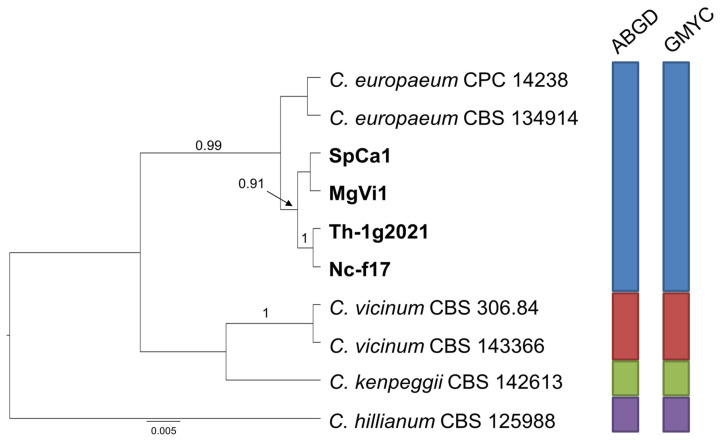
Ultrametric tree phylogeny of group D showing results of sequence-based species delimitation methods. The tree is the result of a Bayesian analysis performed in BEAST on the concatenated ITS, TEF1, ACT dataset. For each node, posterior probabilities (if >0.90) are presented above the branch leading to that node. Results of species delimitation analyses through ABGD and GMYC methods are congruent, as visualized by colored boxes to the right.

**Figure 6 pathogens-10-00369-f006:**
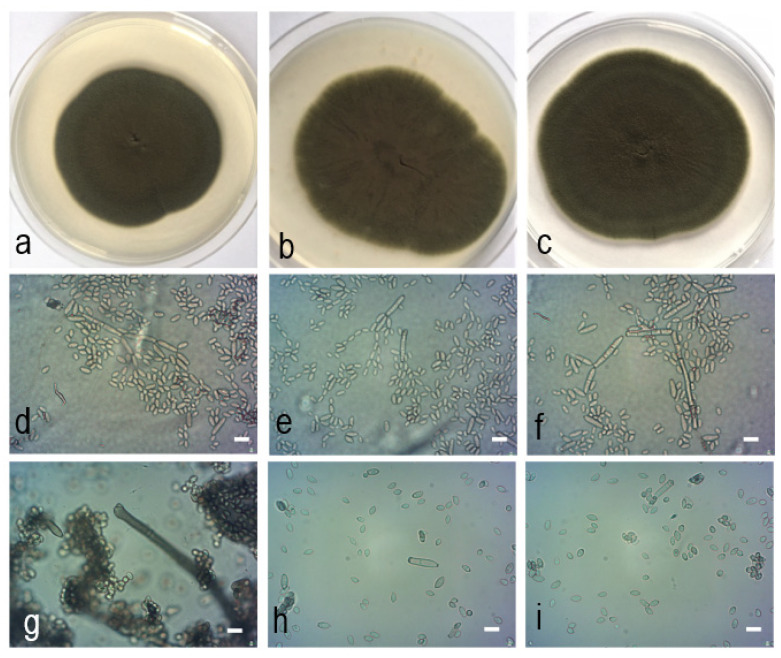
*Cladosporium polonicum* Zimowska & Król *sp. nov. (*isolate Th/lg/2334, holotype). (**a**). Colony on PDA after 14 d; (**b**). colony on OA after 14 d; (**c**). colony on MEA after 14 d; (**d**–**i**). conidiophores and conidial chains; (**e**). tip of conidiophores and numerous conidia; (**f**). cylindrical-oblong, 0−1(−3) septate secondary ramoconidia and conidia; (**g**). tip of a conidiophore with several subdenticulate loci; (**h**–**i**). conidia.—Scale bars = 5 µm.

**Figure 7 pathogens-10-00369-f007:**
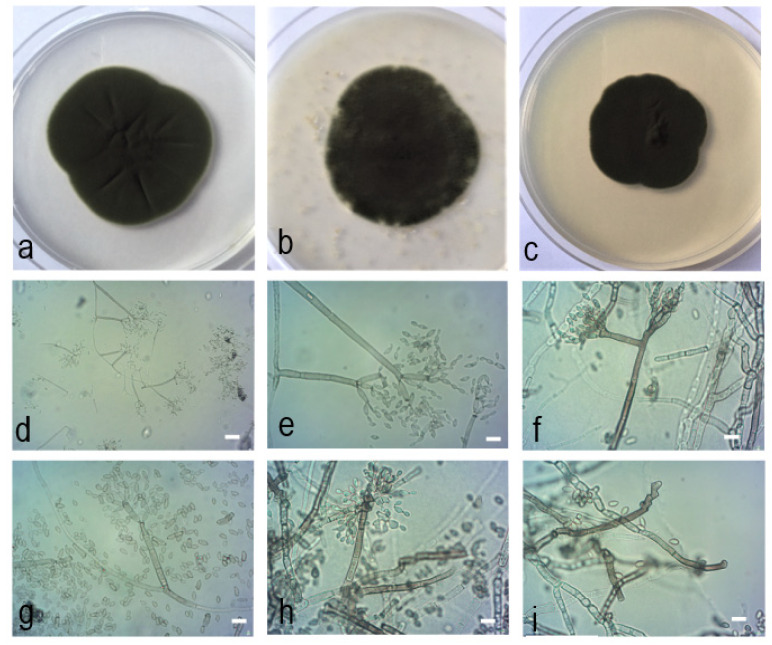
*Cladosporium neapolitanum* Zimowska, Nicoletti & Król *sp. nov.* (isolate MgPo1, holotype). (**a**). Colony on PDA after 14 d; (**b**). colony on OA after 14 d; (**c**). colony on MEA after 14 d; (**d**–**i**). conidiophores and conidial chains; (**d**–**e**). macro- and micronematous conidiophores growing at an angle of 45−90°; (**f**). branched condiophore; (**g**). one septum secondary ramoconidium with broadly truncate base and four apical hila; (**h**–**i**). peculiar conidiogenesis characterized by sympodial proliferation of conidiogenous loci.—Scale bars = 5 µm.

**Table 1 pathogens-10-00369-t001:** List of *Cladosporium* strains from accepted taxa and their corresponding DNA sequences which have been used in the phylogenetic analyses.

Species	Code	Host	Country	ITS	TEF1	ACT
*C. acalyphae*	CBS 125982	*Acalypha australis*	South Korea	HM147994	HM148235	HM148481
*C. alboflavescens*	CBS 140690	bronchoalveolar lavage fluid	United States	LN834420	LN834516	LN834604
*C. angulosum*	CPC 22271	indoor air	United States	MF472918	MF473345	MF473768
*C. angulosum*	CBS 140692	bronchoalveolar lavage fluid	United States	LN834425	LN834521	LN834609
*C. angustisporum*	CBS 125983	*Alloxylon wickhamii*	Australia	HM147995	HM148236	HM148482
*C. angustisporum*	DTO-127-E6	air in bakery	United States	KP701935	KP701812	KP702057
*C. angustiterminale*	CBS 140480	*Banksia grandis*	Australia	KT600379	KT600476	KT600575
*C. anthropophilum*	CBS 117483	-	United States	HM148007	HM148248	HM148494
*C. anthropophilum*	CPC 22393	indoor air	United States	MF472922	MF473349	MF473772
*C. arenosum*	CHFC-EA 566	marine sediment	Antarctica	MN879328	MN890011	MN890008
*C. asperulatum*	CBS 126340	*Protea susannae*	Portugal	HM147998	HM148239	HM148485
*C. asperulatum*	CBS 126339	leaf litter	India	HM147997	HM148238	HM148484
*C. australiense*	DTO-255-F3	bathroom	Netherlands	KP701978	KP701855	KP702100
*C. australiense*	CBS 125984	*Eucalyptus moluccana*	Australia	HM147999	HM148240	HM148486
*C. austroafricanum*	CBS 140481	leaf litter	South Africa	KT600381	KT600478	KT600577
*C. chalastosporoides*	CBS 125985	*Teratosphaeria proteae-arboreae* on *Protea arborea*	South Africa	HM148001	HM148242	HM148488
*C. chasmanthicola*	CBS 142612	*Chasmanthe aethiopica*	South Africa	KY646221	KY646227	KY646224
*C. chubutense*	CBS 124457	*Pinus ponderosa*	Argentina	FJ936158	FJ936161	FJ936165
*C. cladosporioides*	CBS 113739	crested wheat grass	United States	HM148005	HM148246	HM148492
*C. cladosporioides*	CBS 145.35	*Pisum sativum*	Germany	HM148013	HM148254	HM148500
*C. cladosporioides*	CBS 101367	soil	Brazil	HM148002	HM148243	HM148489
*C. cladosporioides*	CBS 112388	indoor air	Germany	HM148003	HM148244	HM148490
*C. cladosporioides*	CPC 15615	wild tree	Mexico	KT600386	KT600483	KT600581
*C. cladosporioides*	CPC 22367	indoor air	United States	MF472941	MF473368	MF473791
*C. cladosporioides*	CPC 14271	unidentified tree	France	HM148045	HM148286	HM148532
*C. cladosporioides*	CPC 15626	wild plant	Mexico	KT600387	KT600484	KT600582
*C. colocasiae*	CBS 386.64	*Colocasia esculenta*	Taiwan	HM148067	HM148310	HM148555
*C. colocasiae*	CBS 119542	*Colocasia esculenta*	Japan	HM148066	HM148309	HM148554
*C. colombiae*	CBS 274.80B	*Cortaderia sp.*	Colombia	FJ936159	FJ936163	FJ936166
*C. crousii*	CBS 140686	bronchoalveolar lavage fluid	United States	LN834431	LN834527	LN834615
*C. cucumerinum*	CBS 174.62	painted floor	United States	HM148076	HM148320	HM148565
*C. cucumerinum*	CBS 174.54	*Cucumis sativus*	Netherlands	HM148075	HM148319	HM148564
*C. delicatulum*	CBS 126342	indoor air	Denmark	HM148079	HM148323	HM148568
*C. delicatulum*	CBS 126344	*Tilia cordata*	Germany	HM148081	HM148325	HM148570
*C. europaeum*	CBS 134914	building material	Denmark	HM148056	HM148298	HM148543
*C. europaeum*	CPC 14238	fruit of *Sambucus nigra*	Netherlands	HM148055	HM148297	HM148542
*C. exasperatum*	CBS 125986	*Eucalyptus tintinnans*	Australia	HM148090	HM148334	HM148579
*C. exile*	CBS 125987	*Phyllactinia guttata* on leaf of *Corylus* sp.	United States	HM148091	HM148335	HM148580
*C. flabelliforme*	CBS 126345	*Melaleuca cajuputi*	Australia	HM148092	HM148336	HM148581
*C. flavovirens*	CBS 140462	toe nail	United States	LN834440	LN834536	LN834624
*C. funiculosum*	CBS 122129	*Vigna umbellata*	Japan	HM148094	HM148338	HM148583
*C. funiculosum*	CBS 122128	*Ficus carica*	Japan	HM148093	HM148337	HM148582
*C. gamsianum*	CBS 125989	*Strelitzia* sp.	South Africa	HM148095	HM148339	HM148584
*C. gamsianum*	CPC 15617	seeds of *Glycine max*	Mexico	KT600392	KT600489	KT600587
*C. globisporum*	CBS 812.96	meat stamp	Sweden	HM148096	HM148340	HM148585
*C. globisporum*	DTO-220-D4	indoor environment	Netherlands	KP701967	KP701844	KP702089
*C. grevilleae*	CBS 114271	leaf of *Grevillea* sp.	Australia	JF770450	JF770472	JF770473
*C. hillianum*	CBS 125988	leaf of *Typha orientalis*	New Zealand	HM148097	HM148341	HM148586
*C. hillianum*	CPC 15458	leaf of *Typha orientalis*	New Zealand	HM148098	HM148342	HM148587
*C. inversicolor*	CBS 401.80	*Triticum aestivum*	Netherlands	HM148101	HM148345	HM148590
*C. inversicolor*	DTO-108-F8	indoor environment	France	KP701908	KP701785	KP702031
*C. ipereniae*	CBS 140483	*Puya* sp.	Chile	KT600394	KT600491	KT600589
*C. ipereniae*	CPC 16855	*Arctostaphylos pallida*	United States	KT600395	KT600492	KT600590
*C. iranicum*	CBS 126346	leaf of *Citrus sinensis*	Iran	HM148110	HM148354	HM148599
*C. kenpeggii*	CBS 142613	leaf of *Passiflora edulis*	Australia	KY646222	KY646228	KY646225
*C. licheniphilum*	CBS 125990	*Physcia* sp.	Germany	HM148111	HM148355	HM148600
*C. longicatenatum*	CBS 140485	unknown plant	Australia	KT600403	KT600500	KT600598
*C. lycoperdinum*	CBS 274.80C	*Puya* sp.	Colombia	HM148114	HM148358	HM148603
*C. lycoperdinum*	CBS 126347	gall of *Apiosporina morbosa* on *Prunus* sp.	Canada	HM148112	HM148601	HM148601
*C. montecillanum*	CBS 140486	pine needles	Mexico	KT600406	KT600504	KT600602
*C. montecillanum*	CPC 15605	*Taraxacum* sp.	Mexico	KT600407	KT600505	KT600603
*C. myrtacaearum*	CBS 126350	*Corymbia foelscheana*	Australia	HM148117	HM148361	HM148606
*C. myrtacaearum*	CBS 126349	*Eucalyptus placita*	Australia	MH863925	HM148360	HM148605
*C. needhamense*	CBS 143359	indoor air sample	United States	MF473142	MF473570	MF473991
*C. neerlandicum*	CBS 143360	archive dust	Netherlands	KP701887	KP701764	KP702010
*C. neopsychrotolerans*	CGMCC3.18031	rhizosphere of *Saussurea involucrata*	China	KX938383	KX938400	KX938366
*C. neopsychrotolerans*	CGMCC3.18032	rhizosphere of *Saussurea involucrata*	China	KX938384	KX938401	KX938367
*C. oxysporum*	CBS 125991	soil	China	HM148118	HM148362	HM148607
*C. oxysporum*	CBS 126351	indoor air	Venezuela	HM148119	HM148363	HM148608
*C. paracladosporioides*	CBS 171.54	-	-	HM148120	HM148364	HM148609
*C. parapenidielloides*	CBS 140487	*Eucalyptus* sp.	Australia	KT600410	KT600508	KT600606
*C. perangustum*	CBS 125996	*Cussonia* sp.	South Africa	HM148121	HM148365	HM148610
*C. perangustum*	CBS 126365	*Phyllactinia guttata* on leaf of *Corylus* sp.	United States	MH863940	HM148367	HM148612
*C. phaenocomae*	CBS 128769	*Phaenocoma prolifera*	South Africa	JF499837	JF499875	JF499881
*C. phaenocomae*	CPC 18221	*Phaenocoma prolifera*	South Africa	JF499838	JF499876	JF499882
*C. phyllactiniicola*	CBS 126354	*Phyllactinia guttata* on leaf of *Corylus* sp.	United States	MH863930	HM148396	HM148641
*C. phyllactiniicola*	CBS 126355	*Phyllactinia guttata* on leaf of *Corylus* sp.	United States	HM148153	HM148397	HM148642
*C. phyllophilum*	CBS 125992	*Taphrina* sp. on *Prunus cerasus*	Germany	HM148154	HM148398	HM148643
*C. phyllophilum*	CPC 13873	*Teratosphaeria proteae-arboreae* on *Protea arborea*	South Africa	HM148155	HM148399	HM148644
*C. pini-ponderosae*	CBS 124456	*Pinus ponderosa*	Argentina	FJ936160	FJ936164	FJ936167
*C. pseudochalastosporoides*	CBS 140490	pine needles	Mexico	KT600415	KT600513	KT600611
*C. pseudocladosporioides*	CBS 125993	air	Netherlands	HM148158	HM148402	HM148647
*C. pseudocladosporioides*	CBS 117153	leaf of *Paeonia* sp.	Germany	HM148157	HM148401	HM148646
*C. ramotenellum*	CPC 14300	building material	Denmark	KT600438	KT600537	KT600635
*C. rectoides*	CBS 125994	*Vitis flexuosa*	South Korea	HM148193	HM148438	HM148683
*C. rectoides*	CBS 126357	*Plectranthus* sp.	South Korea	MH863933	HM148439	HM148684
*C. rugulovarians*	CBS 140495	unidentified Poaceae	Brazil	KT600459	KT600558	KT600656
*C. scabrellum*	CBS 126358	*Ruscus hypoglossum*	Slovenia	HM148195	HM148440	HM148685
*C. silenes*	CBS 109082	*Silene uniflora*	United Kingdom	EF679354	EF679429	EF679506
*C. silenes*	MFLUCC 17-0195	*Vitis vinifera*	China	MG938717	MG938830	MG938682
*C. sinuatum*	CGMCC3.18096	soil	China	KX938385	KX938402	KX938368
*C. sinuatum*	CGMCC3.18097	soil	China	KX938386	KX938403	KX938369
*Cladosporium* sp.	UTHSC DI-13-227	human sputum	United States	LN834422	LN834518	LN834606
*Cladosporium* sp.	UTHSC DI-13-245	toe	United States	LN834429	LN834525	LN834613
*Cladosporium* sp.	UTHSC DI-13-265	bronchoalveolar lavage fluid	United States	LN834435	LN834531	LN834619
*Cladosporium* sp.	UTHSC DI-13-218	bronchoalveolar lavage fluid	United States	LN834418	LN834514	LN834602
*Cladosporium* sp.	UTHSC DI-13-210	human skin	United States	LN834414	LN834510	LN834598
*C. subuliforme*	CBS 126500	*Chamaedorea metallica*	Thailand	HM148196	HM148441	HM148686
*C. subuliforme*	DTO-130-H8	indoor environment	Thailand	KP701938	KP701815	KP702060
*C. tenuissimum*	XCSY3	*Coriandrum sativum*	China	MG873079	MT154184	MT154174
*C. tenuissimum*	CBS 125995	*Lagerstoemia* sp.	United States	HM148197	HM148442	HM148687
*C. tianshanense*	CGMCC3.18033	rhizosphere of *Saussurea involucrata*	China	KX938381	KX938398	KX938364
*C. tianshanense*	CGMCC3.18034	rhizosphere of *Saussurea involucrata*	China	KX938382	KX938399	KX938365
*C. uredinicola*	CPC 5390	*Cronartium fusiforme* on *Quercus nigra*	United States	AY251071	HM148467	HM148712
*C. uwebraunianum*	CBS 143365	indoor air	Netherlands	MF473306	MF473729	MF474156
*C. uwebraunianum*	DTO-305-H9	house dust	New Zealand	MF473307	MF473730	MF474157
*C. varians*	CBS 126361	leaf debris	India	MH863937	HM148469	HM148714
*C. varians*	CBS 126362	*Catalpa bungei*	Russia	HM148224	HM148470	HM148715
*C. verrucocladosporioides*	CBS 126363	*Rhus chinensis*	South Korea	HM148226	HM148472	HM148717
*C. vicinum*	CBS 143366	indoor air	United States	MF473311	MF473734	MF474161
*C. vicinum*	CBS 306.84	uredospore of *Puccinia allii*	United Kingdom	HM148057	HM148299	HM148544
*C. vignae*	CBS 121.25	*Vigna unguiculata*	United States	HM148227	HM148473	HM148718
*C. welwitschiicola*	CPC 18648	*Welwitschia mirabilis*	Namibia	KY646223	KY646229	KY646226
*C. westerdijkiae*	CPC 10150	*Fatoua villosa*	South Korea	HM148062	HM148304	HM148549
*C. westerdijkiae*	CPC 14284	*Triticum* sp.	Germany	HM148065	HM148307	HM148552
*C. xanthocromaticum*	CBS 126364	*Erythrophleum chlorostachys*	Australia	HM148122	HM148366	HM148611
*C. xanthocromaticum*	CPC 22239	indoor air	United States	MF473316	MF473739	MF474166
*C. xylophilum*	CBS 125997	dead wood of *Picea abies*	Russia	HM148230	HM148476	HM148721
*C. xylophilum*	CBS 113749	*Prunus avium*	United States	HM148228	HM148474	HM148719

**Table 2 pathogens-10-00369-t002:** List of *Cladosporium* isolates recovered from galled and non-galled flowers of Lamiaceae which have been considered in the present study, with GenBank codes of the deposited DNA sequences.

Strain	Source	Location	ITS	TEF1	ACT
AjNa1	*Ajuga reptans*—receptacle	Napoli	MK387884	MK416088	MK416045
AcAv2	*Clinopodium nepeta*—achene	Averno	MK387911	MK416115	MK416072
AcAv4	*Clinopodium nepeta*—larva of *A. nepetae*	Averno	MK387888	MK416092	MK416049
AcAv16	*Clinopodium nepeta*—larva of parasitoid	Averno	MK387905	MK416109	MK416066
AcBa1	*Clinopodium nepeta*—larva of *A. nepetae*	Napoli	MK387916	MK416120	MK416077
AcBa2	*Clinopodium nepeta—*gall wall	Napoli	MK387899	MK416103	MK416060
AcBa3	*Clinopodium nepeta—*gall wall	Napoli	MK387917	MK416121	MK416078
AcBa8	*Clinopodium nepeta—*larva of parasitoid	Napoli	MK387906	MK416110	MK416067
AcCe1	*Clinopodium nepeta—*gall wall	Caserta	MK387910	MK416114	MK416071
AcMn6	*Clinopodium nepeta—*gall wall	Montenuovo	MK387914	MK416118	MK416075
AcMt5	*Clinopodium nepeta—*gall wall	Matera	MK387880	MK416084	MK416041
AcMt6	*Clinopodium nepeta—*larva of *A. nepetae*	Matera	MK387883	MK416087	MK416044
AcNa1	*Clinopodium nepeta—*gall wall	Astroni	MK387881	MK416085	MK416042
AcPp1	*Clinopodium nepeta—*gall wall	Pietrapertosa	MK387900	MK416104	MK416061
AcPp2	*Clinopodium nepeta—*receptacle	Pietrapertosa	MK387885	MK416089	MK416046
AcRi7	*Clinopodium nepeta—*receptacle	Rivello	MK387886	MK416090	MK416047
SG8	*Clinopodium nepeta—*gall wall	San Giorgio a Cremano	MK387907	MK416111	MK416068
CL1	*Clinopodium vulgare—*gall wall	Rivello	MK387908	MK416112	MK416069
CL3	*Clinopodium vulgare—*gall wall	Rivello	MK387898	MK416102	MK416059
CL4	*Clinopodium vulgare—*achene	Rivello	MK387904	MK416108	MK416065
S1	*Clinopodium vulgare—*achene	Grunau im Almtal	MK387902	MK416106	MK416063
LaPo1	*Lamiastrum* sp.—receptacle	Pontone	MK387878	MK416082	MK416039
LaNa1	*Lamium album*—receptacle	Napoli	MK387903	MK416107	MK416064
LaVe1	*Lamium bifidum*—receptacle	Ottaviano	MK387879	MK416083	MK416040
LaPo2	*Lamium purpureum*—receptacle	Portici	MK387877	MK416081	MK416038
MfCa2	*Micromeria fruticulosa—*gall wall	Capri	MK387882	MK416086	MK416043
MgPo1	*Micromeria graeca*—receptacle	Pontone	MK387890	MK416094	MK416051
MgLu1	*Micromeria graeca*—ovary	Lucrino	MK387918	MK416122	MK416079
MgLu2	*Micromeria graeca*—receptacle	Lucrino	MK387901	MK416105	MK416062
MgVi1	*Micromeria graeca—*gall wall	Vivara	MK387893	MK416097	MK416054
MgVi2	*Micromeria graeca—*larva of *Asphondylia* sp.	Vivara	MK387887	MK416091	MK416048
MgVi3	*Micromeria graeca*—receptacle	Vivara	MK387892	MK416096	MK416053
Nc/f17	*Nepeta cataria*—receptacle	Konopnica	MK387896	MK416100	MK416057
SpCa1	*Salvia* sp.—receptacle	Capri	MK387891	MK416095	MK416052
ThSC1	*Thymus* sp.—receptacle	Monte Santa Croce	MK387909	MK416113	MK416070
Th/S345	*Thymus vulgaris—*achene	Fajsławice	MK387889	MK416093	MK416050
Th/lg/2015	*Thymus vulgaris—*gall wall	Fajsławice	MK387912	MK416116	MK416073
Th/lg/2031	*Thymus vulgaris—*gall wall	Fajsławice	MK387897	MK416101	MK416058
Th/lg/2334	*Thymus vulgaris—*gall wall	Fajsławice	MK387894	MK416098	MK416055
Th/k/258	*Thymus vulgaris—*receptacle	Fajsławice	MK387895	MK416099	MK416056

**Table 3 pathogens-10-00369-t003:** Morphological characteristics of *Cladosporium species novae* described in this study.

Strains	Conidiophores (µm)	Ramoconidia (µm)	Secondary ^1^ Ramoconidia (µm)	Intercalary Conidia ^1^ (µm)	Conidia ^1^ (µm)	Colony Diameter ^2^ after 14 Days (mm)
Th/lg/2334	(22−)70−130 × 2−3.6	12.9−36 × 2.5−4.1, 0−1 septate	9.5−17.1 × 2.4−4	6.4−11 × 2−3.5	3−5.8(−6) × (1.5−)2−2.8	PDA: 63 Malt-extract agar (MEA): 58 Oatmeal agar (OA): 53
Th/k/258	(25−)42.7−151 × 2.4−5.1	14.3−39.8 × 2.4−5.2, 0−1 septate	7.9−23.2 × 2.6−4	5.6−8.8 × 2−3.9	3.8−5.6 × (1.5−)2−3	PDA: 70 MEA: 53 OA: 60
*Cladosporium pseudocladosporioides* [3]	15−155 × 2−4	19−48 × 3−4, 0−2(−3) septate	16.1 × 2.9	8.8 × 2.6	4.1 × 2.1	PDA: 65−78 MEA: 52−75 OA: 55−73
MgPo1	(28.1−)44.4−142.5 × (2.1−)2.5−3.9(−4.5)	10.1−20.1 × 2.2−3.7 (−4.3)	(7.1−)8.3−14.6 × 2.1−3.1, 2−4 apical hila	6.1−9.5 × 2−2.9	(2.1−)2.4−4.9(−5.1) × (1.7−)2.1−2.5(−2.8)	PDA: 47 MEA: 37 OA: 46
MgVi3	(57−)68−126.5 × (1.9−)2.4−4.2	11.5−22.2 × 2.4−3.5(−3.9)	(5.7−)7.4−16.6 × 1.8−2.7 2−4 apical hila	5.8−10.5 × 2−3.3	(1.4−)2.2−4.3 × (1.3−)1.6−2.5(−2.8)	PDA: 47 MEA: 40 OA: 45
*Cladosporium xylophilum* [3]	155(−190) × 2−4(−5)	19-35 × 2.5-3	14.5(±5.1) × 3.1(±0.5), up to 6(−9) apical hila	7.7(±2.2) × 2.6(±0.3)	3.9(±0.9) × 2.3(±0.3)	PDA: 52−74 MEA: 47−74 OA: 47−58

^1^ Average of 50 measurements. ^2^ Average of three replicates.

## Data Availability

No new data were created or analyzed in this study. Data sharing is not applicable to this article.

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
