# Peer review of "New *Cladosporium* Species from Normal and Galled Flowers of Lamiaceae"

_pathogens, 2021, doi:10.3390/pathogens10030369_

Round 1
Reviewer 1 Report
Evaluation and comments to the manuscript ID pathogens-1143776, and entitled “New Cladosporium Species from Normal and Galled Flowers of Lamiaceae”.
Authors: Beata Zimowska et al.
Overall, the manuscript is written in correct scientific language. I did not notice any major factual or editorial errors at work, and I do not mention any minor errors, because they are in some sense something normal in this type of work and they do not affect the general perception of work.
In my opinion, the experimental design and data analysis are appropriate and the introduction is correct. I think that we can always do something better, but this manuscript is on a good level. However, a few things need to be corrected in the manuscript:
1) It is necessary to correct minor omissions, for example, unnecessary underlining of data in Table 3.
2) improve readability Fig. 1
3) Table 1 – co is this table so important that it should be in the main body of the manuscript? Maybe put it as supplementary materials. I leave the decision to the author.
Author Response
Thank you very much for your positive opinion and suggestions for improving our manuscript. Concerning specific comments, we observe that:
1) It is necessary to correct minor omissions, for example, unnecessary underlining of data in Table 3.
In Table 3, underlining has been removed in 2nd and 3rd line of 3rd column;
2) improve readability Fig. 1
Fig. 1 was adapted to page size. However, we recommend adjustment to the most convenient size to be done at the proof stage;
3) Table 1 – co is this table so important that it should be in the main body of the manuscript? Maybe put it as supplementary materials. I leave the decision to the author.
Table 1 reports the list of reference strains used in phylogenetic analysis. Actually, this information is very important, so it must integrate the main text.
Reviewer 2 Report
New Cladosporium Species from Normal and Galled Flowers of Lamiaceae.
This is an investigation into the presence of fungi of the genus Cladosporium within the galls produced on various Lamiaceae by midges of the genus Asphondylia, which led to the discovery of two undescribed species. The manuscript is well structured and the arguments are robustly supported by molecular analyzes and descriptions. It seems to me clearly and appropriately written. however I suggest a few slight changes.
186-188 Table 3. Morphological characteristics of Cladosporium species novae described in this study. Main differences with the most closely related species are underlined.
And sizes in the descriptions
L 207, 2012, 2016, 220, 222, 268,274,279,284.
In Table 3 and in the descriptions of the two new species, the minimum and maximum sizes and averages are given. For a taxonomic work, at least in the descriptions and at least limited to the averages, the standard deviation SD values should be given.
Also in Table 3 the values that are underlined as different between the different species are a deduction of the AA as it seems that they have not been subjected to any statistical verification that has indicated their significance. I understand that most of the comparison values reported are taken from the literature and that sometimes for that small range it is difficult to obtain reliable measurements even at maximum magnification, but at least there are explanations at the M&M level and a minimum of motivations in the discussion.
191-192 Cladosporium polonicum Zimowska & Król sp. nov. − MycoBank MBXXXXXX; Fig. 6.
Differs from C. pseudocladosporioides in forming slightly shorter, 0−1 septate ramoconidia and shorter secondary ramoconidia.
In the diagnosis it is better to first frame the new taxon morphologically and then give the differences, so I suggest changing the opening words:
‘Similar to Cladosporium pseudocladosporioides from which it differs in …’
245 Fig. 6. Cladosporium polonicum sp. nov.
Better ‘Cladosporium polonicum Zimowska & Król sp. nov.’
250-252 Cladosporium neapolitanum Zimowska, Nicoletti & Król sp. nov. − MycoBank 249 MBXXXXXX; Fig. 7.
Differs from C. xylophilum in forming shorter conidiophores, shorter ramoconidia and secondary ramoconidia, and for a lower number of hila at the apex of secondary ramoconidia.
‘Similar to Cladosporium xylophilum from which it differs in …’
307 Fig. 7. Cladosporium neapolitanum sp. nov.
Better ‘Cladosporium neapolitanum Zimowska, Nicoletti & Król sp. nov.’
Author Response
Thank you very much for your positive opinion and suggestions for improving our manuscript. Concerning specific comments, we observe that:
186-188 Table 3. Morphological characteristics of Cladosporium species novae described in this study. Main differences with the most closely related species are underlined. And sizes in the descriptions L 207, 2012, 2016, 220, 222, 268,274,279,284. In Table 3 and in the descriptions of the two new species, the minimum and maximum sizes and averages are given. For a taxonomic work, at least in the descriptions and at least limited to the averages, the standard deviation SD values should be given.
Morphological descriptions usually do not aim to assess significance in statistical terms, also considering that a comparison with known species by growing them in exactly the same conditions in not possible. Hence, in our case we could still calculate standard deviations with reference to our original measurements, but could not analyze them in comparison with the most closely related species.
Also in Table 3 the values that are underlined as different between the different species are a deduction of the AA as it seems that they have not been subjected to any statistical verification that has indicated their significance. I understand that most of the comparison values reported are taken from the literature and that sometimes for that small range it is difficult to obtain reliable measurements even at maximum magnification, but at least there are explanations at the M&M level and a minimum of motivations in the discussion.
In our discussion we stated that ‘…it is clear that in the absence of reliable morphological characters the use of sequence-based statistical methods able to assess the significant phylogenetic distances is to be recommended in view of a correct classification, as well as to avoid the accumulation of misleading identifications…’. Indeed, the phylogenetic approach enables to support species discrimination in statistical terms, which overcomes uncertainties due to the lack of clear morphological differences.
191-192 Cladosporium polonicum Zimowska & Król sp. nov. − MycoBank MBXXXXXX; Fig. 6.
Differs from C. pseudocladosporioides in forming slightly shorter, 0−1 septate ramoconidia and shorter secondary ramoconidia.
In the diagnosis it is better to first frame the new taxon morphologically and then give the differences, so I suggest changing the opening words:
‘Similar to Cladosporium pseudocladosporioides from which it differs in …’
245 Fig. 6. Cladosporium polonicum sp. nov.
Better ‘Cladosporium polonicum Zimowska & Król sp. nov.’
250-252 Cladosporium neapolitanum Zimowska, Nicoletti & Król sp. nov. − MycoBank 249 MBXXXXXX; Fig. 7.
Differs from C. xylophilum in forming shorter conidiophores, shorter ramoconidia and secondary ramoconidia, and for a lower number of hila at the apex of secondary ramoconidia.
‘Similar to Cladosporium xylophilum from which it differs in …’
307 Fig. 7. Cladosporium neapolitanum sp. nov.
Better ‘Cladosporium neapolitanum Zimowska, Nicoletti & Król sp. nov.’
All these corrections have been done. Moreover, the official Mycobank numbers have been added.